# The Fairness of Risk Scores Beyond Classification: Bipartite Ranking and the xAUC Metric

**Nathan Kallus**
Cornell University
New York, NY
kallus@cornell.edu

**Angela Zhou**
Cornell University
New York, NY
az434@cornell.edu

## Abstract

Where machine-learned predictive risk scores inform high-stakes decisions, such as bail and sentencing in criminal justice, fairness has been a serious concern. Recent work has characterized the disparate impact that such risk scores can have when used for a binary classification task. This may not account, however, for the more diverse downstream uses of risk scores and their non-binary nature. To better account for this, in this paper, we investigate the fairness of predictive risk scores from the point of view of a bipartite ranking task, where one seeks to rank positive examples higher than negative ones. We introduce the xAUC disparity as a metric to assess the disparate impact of risk scores and define it as the difference in the probabilities of ranking a random positive example from one protected group above a negative one from another group and vice versa. We provide a decomposition of bipartite ranking loss into components that involve the discrepancy and components that involve pure predictive ability within each group. We use xAUC analysis to audit predictive risk scores for recidivism prediction, income prediction, and cardiac arrest prediction, where it describes disparities that are not evident from simply comparing within-group predictive performance.

## 1 Introduction

Predictive risk scores support decision-making in high-stakes settings such as bail sentencing in the criminal justice system, triage and preventive care in healthcare, and lending decisions in the credit industry [2, 38]. In these areas where predictive errors can significantly impact individuals involved, studies of fairness in machine learning have analyzed the possible disparate impact introduced by predictive risk scores primarily in a *binary classification setting*: if predictions determine whether or not someone is detained pre-trial, is admitted into critical care, or is extended a loan. But the "human in the loop" with risk assessment tools often has recourse to make decisions about extent, intensity, or prioritization of resources. That is, in practice, predictive risk scores are used to provide informative rank-orderings of individuals with binary outcomes in the following settings:

(1) In criminal justice, the "risk-needs-responsivity" model emphasizes matching the level of social service interventions to the specific individual's risk of re-offending [3, 6].

(2) In healthcare and other clinical decision-making settings, risk scores are used as decision aids for prevention of chronic disease or triage of health resources, where a variety of interventional resource intensities are available; however, the prediction quality of individual conditional probability estimates can be poor [9, 28, 38, 39].

(3) In credit, predictions of default risk affect not only loan acceptance/rejection decisions, but also risk-based setting of interest rates. Fuster et al. [22] embed machine-learned credit scores in an economic pricing model which suggests negative economic welfare impacts on Black and Hispanic borrowers.

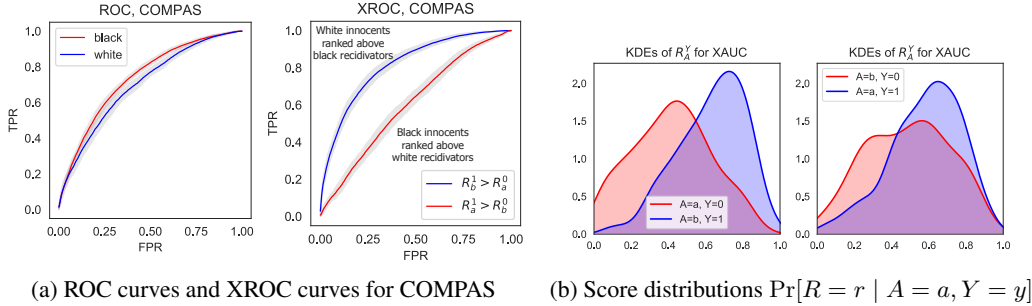

(a) ROC curves and XROC curves for COMPAS  (b) Score distributions $\Pr[R = r \mid A = a, Y = y]$

Figure 1: Analysis of xAUC disparities for the COMPAS Violent Recidivism Prediction dataset

(4) In municipal services, predictive analytics tools have been used to direct resources for maintenance, repair, or inspection by prioritizing or ranking by risk of failure or contamination [12, 40]. Proposals to use new data sources such as 311 data, which incur the self-selection bias of citizen complaints, may introduce inequities in resource allocation [32].

We describe how the problem of *bipartite ranking*, that of finding a good ranking function that ranks positively labeled examples above negative examples, better encapsulates how predictive risk scores are used in practice to rank individual units, and how a new metric we propose, xAUC, can assess ranking disparities.

Most previous work on fairness in machine learning has emphasized disparate impact in terms of confusion matrix metrics such as true positive rates and false positive rates and other desiderata, such as probability calibration of risk scores. Due in part to inherent trade-offs between these performance criteria, some have recommended to retain *unadjusted* risk scores that achieve good calibration, rather than adjusting for parity across groups, in order to retain as much information as possible and allow human experts to make the final decision [10, 14, 15, 27]. At the same time, group-level discrepancies in the *prediction loss* of risk scores, relative to the true Bayes-optimal score, are not observable, since only binary outcomes are observed.

In particular, our bipartite ranking-based perspective reconciles a gap between the differing arguments made by ProPublica and Equivant (then Northpointe) regarding the potential bias or disparate impact of the COMPAS recidivism tool. Equivant levies within-group AUC parity ("accuracy equity") (among other desiderata such as calibration and predictive parity) to claim fairness of the risk scores in response to ProPublica's allegations of bias due to true positive rate/false positive rate disparities for the Low/Not Low risk labels [2, 19]. Our xAUC metric, which measures the probability of positive-instance members of one group being misranked below negative-instance members of another group, and vice-versa, highlights that within-group comparison of AUC discrepancies does not summarize accuracy inequity. We illustrate this in Fig. 1 for a risk score learned from COMPAS data: xAUC disparities reflect disparate misranking risk faced by positive-label individual of either class.

In this paper, we propose and study the cross-ROC curve and the corresponding xAUC metric for auditing disparities induced by a predictive risk score, as they are used in broader contexts to inform resource allocation. We relate the xAUC metric to different group- and outcome-based decompositions of a *bipartite ranking* loss, and assess the resulting metrics on datasets where fairness has been of concern.

## 2   Related Work

Our analysis of fairness properties of risk scores in this work is most closely related to the study of "disparate impact" in machine learning, which focuses on disparities in the *outcomes* of a process across protected classes, without racial animus [4]. Many previous approaches have considered formalizations via error rate metrics of the confusion matrix in a binary classification setting [5, 25, 29, 35, 44]. By now, a panoply of fairness metrics have been studied for binary classification in order to assess group-level disparities in confusion matrix-based metrics. Proposals for error rate balance assess or try to equalize true positive rates and/or false positive rates, error rates measured conditional on the *true* outcome, emphasizing the equitable treatment of those who actually are of the outcome type of interest [25, 44]. Alternatively, one might assess the negative/positive predictive

value (NPV/PPV) error rates conditional on the thresholded *model prediction* [13]. In missing-data settings, these metrics can be partially identified to support fairness assessments [11, 30, 30].

The predominant criterion used for assessing fairness of *risk scores*, outside of a binary classification setting, is that of calibration. Group-wise calibration requires that $\Pr[Y = 1 \mid R = r, A = a] = \Pr[Y = 1 \mid R = r, A = b] = r$, as in [13]. The impossibilities of satisfying notions of error rate balance and calibration simultaneously have been discussed in [13, 31]. Liu et al. [33] show that group calibration is a byproduct of unconstrained empirical risk minimization, and therefore is not a restrictive notion of fairness. Hebert-Johnson et al. [26] note the critique that group calibration does not restrict the *variance* of a risk score as an unbiased estimator of the Bayes-optimal score.

Other work has considered fairness in ranking settings specifically, with particular attention to applications in information retrieval, such as questions of fair representation in search engine results. Yang and Stoyanovich [43] assess statistical parity at discrete cut-points of a ranking, incorporating position bias inspired by normalized discounted cumulative gain (nDCG) metrics. Celis et al. [8] consider the question of fairness in rankings, where fairness is considered as constraints on diversity of group membership in the top $k$ rankings, for any choice of $k$. Singh and Joachims [41] consider fairness of exposure in rankings under known relevance scores and propose an algorithmic framework that produces probabilistic rankings satisfying fairness constraints in expectation on exposure, under a position bias model. We focus instead on the bipartite ranking setting, where the area under the curve (AUC) loss emphasizes ranking quality on the entire distribution, whereas other ranking metrics such as nDCG or top-k metrics emphasize only a portion of the distribution.

The problem of bipartite ranking is related to, but distinct from, binary classification [1, 20, 36]; see [16, 34] for more information. While the bipartite ranking induced by the Bayes-optimal score is analogously Bayes-risk optimal for bipartite ranking (*e.g.*, [34]), in general, a probability-calibrated classifier is not optimizing for the bipartite ranking loss. Cortes and Mohri [16] observe that AUC may vary widely for the same error rate, and that algorithms designed to globally optimize the AUC perform better than optimizing surrogates of the AUC or error rate. Narasimhan and Agarwal [37] study transfer regret bounds between the related problems of binary classification, bipartite ranking, and outcome-probability estimation.

## 3 Problem Setup and Notation

We suppose we have data $(X, A, Y)$ on features $X \in \mathcal{X}$, sensitive attribute $A \in \mathcal{A}$, and binary labeled outcome $Y \in \{0, 1\}$. We are interested in assessing the downstream impacts of a predictive risk score $R : \mathcal{X} \times \mathcal{A} \to \mathbb{R}$, which may or may not access the sensitive attribute. When these risk scores represent an estimated conditional probability of positive label, $R : \mathcal{X} \times \mathcal{A} \to [0, 1]$. For brevity, we also let $R = R(X, A)$ be the random variable corresponding to an individual's risk score. We generally use the conventions that $Y = 1$ is associated with opportunity or benefit for the individual (*e.g.*, freedom from suspicion of recidivism, creditworthiness) and that when discussing two groups, $A = a$ and $A = b$, the group $A = a$ might be a historically disadvantaged group.

Let the conditional cumulative distribution function of the learned score $R$ evaluated at a threshold $\theta$ given label and attribute be denoted by

$$F_y^a(\theta) = \Pr[R \le \theta \mid Y = y, A = a].$$

We let $G_y^a = 1 - F_y^a$ denote the complement of $F_y^a$. We drop the $a$ subscript to refer to the whole population: $F_y(\theta) = \Pr[R \le \theta \mid Y = y]$. Thresholding the score yields a binary classifier, $\hat{Y}_\theta = \mathbb{I}[R \ge \theta]$. The classifier's true negative rate (TNR) is $F_0(\theta)$, its false positive rate (FPR) is $G_0(\theta)$, its false negative rate (FNR) is $F_1(\theta)$, and its true positive rate (TPR) is $G_1(\theta)$. Given a risk score, the choice of optimal threshold for a binary classifier depends on the differing costs of false positive and false negatives. We might expect cost ratios of false positives and false negatives to differ if we consider the use of risk scores to direct punitive measures or to direct interventional resources.

In the setting of bipartite ranking, the data comprises of a pool of positive labeled examples, $S_+ = \{X_i\}_{i \in [m]}$, drawn i.i.d. according to a distribution $X_+ \sim D_+$, and negative labeled examples $S_- = \{X_i'\}_{i \in [n]}$ drawn according to a distribution $X_- \sim D_-$ [36]. The rank order may be determined by a score function $s(X)$, which achieves empirical bipartite ranking error $\frac{1}{mn} \sum_{i=1}^{m} \sum_{j=1}^{n} \mathbb{I}[s(X_i) < s(X_j')]$. The area under the receiver operating characteristic (ROC) curve (AUC), a common (reward) objective for bipartite ranking is often used as a metric describing the quality of a predictive score,

independently of the final threshold used to implement a classifier, and is invariant to different base rates of the outcomes. The ROC curve plots $G^0(\theta)$ on the x-axis with $G^1(\theta)$ on the y-axis as we vary $\theta$ over the space of various decision thresholds. The AUC is the area under the ROC curve, *i.e.*,

$$\text{AUC} = \int_0^1 G_1(G_0^{-1}(v))dv$$

An AUC of $\frac{1}{2}$ corresponds to a completely random classifier; therefore, the difference from $\frac{1}{2}$ serves as a metric for the diagnostic quality of a predictive score. We recall the probabilistic interpretation of AUC that it is the probability that a randomly drawn example from the positive class is correctly ranked by the score $R$ above a randomly drawn score from the negative class [24]. Let $R_1$ be drawn from $R \mid Y = 1$ and $R_0$ be drawn from $R \mid Y = 0$ independently. Then $\text{AUC} = \Pr[R_1 > R_0]$.

## 4 The Cross-ROC (xROC) and Cross-Area Under the Curve (xAUC)

We introduce the cross-ROC curve and the cross-area under the curve metric xAUC that summarize group-level disparities in *misranking* errors induced by a score function $R(X, A)$.

**Definition 1** (Cross-Receiver Operating Characteristic curve (xROC))**.**
$$\text{xROC}(\theta; R, a, b) = (\Pr[R > \theta \mid A = b, Y = 0], \Pr[R > \theta \mid A = a, Y = 1])$$

The $\text{xROC}^{a,b}$ curve parametrically plots $\text{xROC}(\theta; R, a, b)$ over the space of thresholds $\theta \in \mathbb{R}$, generating the curve of TPR of group $a$ on the y-axis vs. the FPR of group $b$ on the x-axis. We define the $\text{xAUC}(a, b)$ metric as the area under the $\text{xROC}^{a,b}$ curve. Analogous to the usual AUC, we provide a probabilistic interpretation of the xAUC metric as the probability of correctly ranking a positive instance of group $a$ above a negative instance of group $a$ under the corresponding outcome- and class-conditional distributions of the score.

**Definition 2** (xAUC)**.**

$$\text{xAUC}(a, b) = \int_0^1 G_1^a((G_0^b)^{-1}(v))dv = \Pr[R_1^a > R_0^b]$$

where $R_1^a$ is drawn from $R \mid Y = 1, A = a$ and $R_0^b$ is drawn from $R \mid Y = 0, A = b$ independently. For brevity, henceforth, $R_y^a$ is taken to be drawn from $R \mid Y = y, A = a$ and independently of any other such variable. We also drop the superscript to denote omitting the conditioning on sensitive attribute (*e.g.*, $R_y$).

The xAUC accuracy metrics for a binary sensitive attribute measure the probability that a randomly chosen unit from the "positive" group $Y = 1$ in group $a$, is ranked higher than a randomly chosen unit from the "negative" group, $Y = 0$ in group $b$, under the corresponding group- and outcome-conditional distributions of scores $R_a^y$. We let $\text{AUC}^a$ denote the *within-group* AUC for group $A = a$, $\Pr[R_1^a > R_0^a]$.

If the difference between these metrics, the xAUC disparity
$$\Delta\text{xAUC} = \Pr[R_1^a > R_0^b] - \Pr[R_1^b > R_0^a] = \Pr[R_1^b \leq R_0^a] - \Pr[R_1^a \leq R_0^b]$$
is substantial and positive, then we might consider group $b$ to be systematically "disadvantaged" and $a$ to be "advantaged" when $Y = 0$ is a negative or harmful label or is associated with punitive measures, as in the recidivism predication case. Conversely, we have the opposite interpretation if $Y = 0$ is a positive label associated with greater beneficial resources. Similarly, since $\Delta\text{xAUC}$ is anti-symmetric in $a, b$, negative values are also interpreted in the converse.

When higher scores are associated with opportunity or additional benefits and resources, as in the recidivism predication case, a positive $\Delta\text{xAUC}$ means group $a$ either gains by correctly having its deserving members correctly ranked above the non-deserving members of group $b$ and/or by having its non-deserving members incorrectly ranked above the deserving members of group $b$; and symmetrically, group $b$ loses in the same way. The magnitude of the disparity $\Delta\text{xAUC}$ describes the misranking disparities incurred under this predictive score, while the magnitude of the xAUC measures the particular across-subgroup rank-accuracies.

Computing the xAUC is simple: one simply computes the sample statistic, $\frac{1}{n_0^b n_1^a} \sum_{i:\ A_i=a,\ Y_i=1} \sum_{j:\ A_j=b,\ Y_j=0} \mathbb{I}[R(X_i) > R(X_j)]$. Algorithmic routines for computing the AUC quickly by a sorting routine can be directly used to compute the xAUCs. Asymptotically exact confidence intervals are available, as shown in DeLong et al. [17], using the generalized U-statistic property of this estimator.

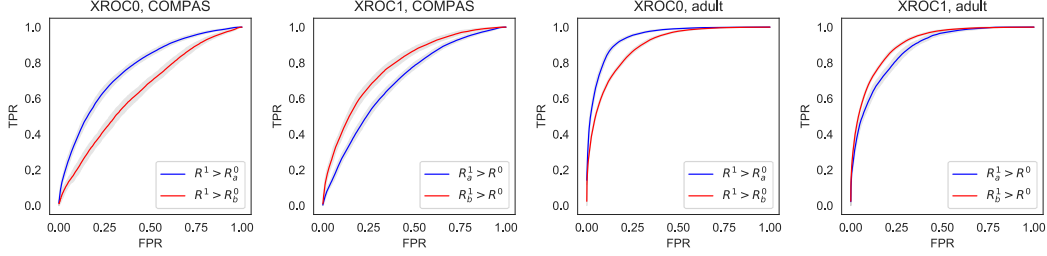

(a) COMPAS: $a = $ Black, $b = $ White, $0 = $ Recidivate    (b) Adult: $a = $ Black, $b = $ White, $0 = $ Low income

Figure 2: Balanced xROC curves for COMPAS and Adult datasets

Table 1: Ranking error metrics (AUC, xAUC, Brier scores for calibration) for different datasets. We include standard errors in Table 2 of the appendix.

| | | COMPAS | | Framingham | | German | | Adult | |
|---|---|---|---|---|---|---|---|---|---|
| | $A = $ | Black | White | Non-F. | Female | $< 25$ | $\geq 25$ | Black | White |
| **Logistic Reg.** | AUC | 0.737 | 0.701 | 0.768 | 0.768 | 0.726 | 0.788 | 0.923 | 0.898 |
| | Brier | 0.208 | 0.21 | 0.201 | 0.166 | 0.211 | 0.158 | 0.075 | 0.111 |
| | XAUC | 0.604 | 0.813 | 0.795 | 0.737 | 0.708 | 0.802 | 0.865 | 0.944 |
| | $\text{XAUC}^1$ | 0.698 | 0.781 | 0.785 | 0.756 | 0.712 | 0.791 | 0.874 | 0.905 |
| | $\text{XAUC}^0$ | 0.766 | 0.641 | 0.755 | 0.783 | 0.79 | 0.775 | 0.943 | 0.895 |
| **RankBoost cal.** | AUC | 0.745 | 0.703 | 0.789 | 0.797 | 0.704 | 0.796 | 0.924 | 0.899 |
| | Brier | 0.206 | 0.21 | 0.182 | 0.15 | 0.22 | 0.158 | 0.074 | 0.109 |
| | XAUC | 0.599 | 0.827 | 0.822 | 0.761 | 0.714 | 0.788 | 0.875 | 0.941 |
| | $\text{XAUC}^1$ | 0.702 | 0.79 | 0.809 | 0.783 | 0.711 | 0.793 | 0.882 | 0.906 |
| | $\text{XAUC}^0$ | 0.776 | 0.638 | 0.777 | 0.811 | 0.774 | 0.783 | 0.939 | 0.897 |

**Variants of the xAUC metric**   We can decompose AUC differently and assess different variants of the xAUC:

**Definition 3** (Balanced xAUC).

$$\text{xAUC}_0(a) = \Pr[R_1 > R_0^a], \ \text{xAUC}^0(b) = \Pr[R_1 > R_0^b]$$
$$\text{xAUC}^1(a) = \Pr[R_1^a > R_0], \ \text{xAUC}^1(b) = \Pr[R_1^b > R_0]$$

These xAUC disparities compare misranking error faced by individuals from either group, conditional on a specific outcome: $\text{xAUC}_0(a) - \text{xAUC}^0(b)$ compares the ranking accuracy faced by those of the negative class $Y = 0$ across groups, and $\text{xAUC}^1(a) - \text{xAUC}^1(b)$ analogously compares those of the positive class $Y = 1$. The following proposition shows how the population AUC decomposes as weighted combinations of the xAUC and within-class AUCs, or the balanced decompositions $\text{xAUC}_1$ or $\text{xAUC}_0$, weighted by the outcome-conditional class probabilities.

**Proposition 1** (xAUC metrics as decompositions of AUC).

$$\text{AUC} = \Pr[R_1 > R_0] = \sum_{b' \in \mathcal{A}} \Pr[A = b' \mid Y = 0] \cdot \sum_{a' \in \mathcal{A}} \Pr[A = a' \mid Y = 1] \Pr[R_1^{a'} > R_0^{b'}]$$
$$= \sum_{a' \in \mathcal{A}} \Pr[A = a' \mid Y = 1] \Pr[R_1^{a'} > R_0] = \sum_{a' \in \mathcal{A}} \Pr[A = a' \mid Y = 0] \Pr[R_1 > R_0^{a'}]$$

## 5 Assessing xAUC

### 5.1 COMPAS Example

In Fig. 1, we revisit the COMPAS data and assess our xROC and xAUC curves to illustrate ranking disparities that may be induced by risk scores learned from this data. The COMPAS dataset is of size

$n = 6167, p = 402$, where sensitive attribute is race, with $A = a, b$ for black and white, respectively. We define the outcome $Y = 1$ for non-recidivism within 2 years and $Y = 0$ for violent recidivism. Covariates include information on number of prior arrests and age; we follow the pre-processing of Friedler et al. [21].

We first train a logistic regression model on the original covariate data (we do not use the decile scores directly in order to do a more fine-grained analysis), using a 70%, 30% train-test split and evaluating metrics on the out-of-sample test set. In Table 1, we report the group-level AUC and the Brier [7] scores (summarizing calibration), and our xAUC metrics. The xAUC for column $A = a$ is xAUC$(a, b)$, for column $A = b$ it is xAUC$(b, a)$, and for column $A = a$, xAUC$^y$ is xAUC$^y(a)$. The Brier score for a probabilistic prediction of a binary outcome is $\frac{1}{n} \sum_i^n (R(X_i) - Y_i)^2$. The score is overall well-calibrated (as well as calibrated by group), consistent with analyses elsewhere [13, 19].

We also report the metrics from using a bipartite ranking algorithm, Bipartite Rankboost of Freund et al. [20] and calibrating the resulting ranking score by Platt Scaling, displaying the results as "RankBoost cal." We observe essentially similar performance across these metrics, suggesting that the behavior of xAUC disparities is independent of model specification or complexity; and that methods which directly optimize the population AUC error may still incur these group-level error disparities.

In Fig. 1a, we plot ROC curves and our xROC curves, displaying the averaged ROC curve (interpolated to a fine grid of FPR values) over 50 sampled train-test splits, with 1 standard error bar shaded in gray (computed by the method of [17]). We include standard errors for xAUC metrics in Table 2 of the appendix. While a simple *within-group* AUC comparison suggests that the score is overall more accurate for blacks – in fact, the AUC is slightly higher for the black population with AUC$^a = 0.737$ and AUC$^b = 0.701$ – computing our xROC curve and xAUC metric shows that blacks would be disadvantaged by misranking errors. The cross-group accuracy xAUC$(a, b) = 0.604$ is significantly lower than xAUC$(b, a) = 0.813$: black innocents are nearly indistinguishable from actually guilty whites. This $\Delta$ xAUC gap of $-0.21$ is precisely the **cross-group accuracy inequity** that simply comparing *within-group AUC* does not capture. When we plot kernel density estimates of the score distributions in Fig. 1b from a representative training-test split, we see that indeed the distribution of scores for black innocents $\Pr[R = r \mid A = a, Y = 0]$ has significant overlap with the distribution of scores for white innocents.

**Assessing balanced xROC:** In Fig. 2, we compare the xROC$_0(a)$, xROC$_0(b)$ curves with the xROC$_1(a)$, xROC$_1(b)$ curves for the COMPAS data. The relative magnitude of $\Delta$ xAUC$_1$ and $\Delta$ xAUC$_0$ provides insight on whether the burden of the xAUC disparity falls on those who are innocent or guilty. Here, since the $\Delta$ xAUC$_0$ disparity is larger in absolute terms, it seems that misranking errors result in inordinate benefit of the doubt in the errors of distinguishing risky whites ($Y = 0$) from innocent individuals, rather than disparities arising from distinguishing innocent members of either group from generally guilty individuals.

## 5.2  Assessing xAUC on Other Datasets

Additionally in Fig. 3 and Table. 1, we evaluate these metrics on multiple datasets where fairness may be of concern, including risk scores learnt on the Framingham study, the German credit dataset, and the Adult income prediction dataset (we use logistic regression as well as calibrated bipartite RankBoost) [18, 42]. For the Framingham dataset (cardiac arrest risk scores), $n = 4658, p = 7$ with sensitive attribute of gender, $A = a$ for non-female and $A = b$ for female. $Y = 1$ denotes 10-year coronary heart disease (CHD) incidence. Fairness considerations might arise if predictions of likelier mortality are associated with greater resources for preventive care or triage. The German credit dataset is of size $n = 1000, p = 57$, where the sensitive attribute is age with $A = a, b$ for age $< 25$, age $> 25$. Creditworthiness (non-default) is denoted by $Y = 1$, and default by $Y = 0$. The "Adult" income dataset is of size $n = 30162, p = 98$, sensitive attribute, $A = a, b$ for black and white. We use the dichotomized outcome $Y = 1$ for high income $> 50$k, $Y = 0$ for low income $< 50$k.

Overall, Fig. 3 shows that these xAUC disparities persist, though the disparities are largest for the COMPAS and the large Adult dataset. For the Adult dataset this disparity could result in the misranking of poor whites above wealthy blacks; this could be interpreted as possibly inequitable withholding of economic opportunity from actually-high-income blacks. The additional datasets also display different phenomena regarding the score distributions and xROC$_0$, xROC$_1$ comparisons, which we include in Fig. 5 of the Appendix.

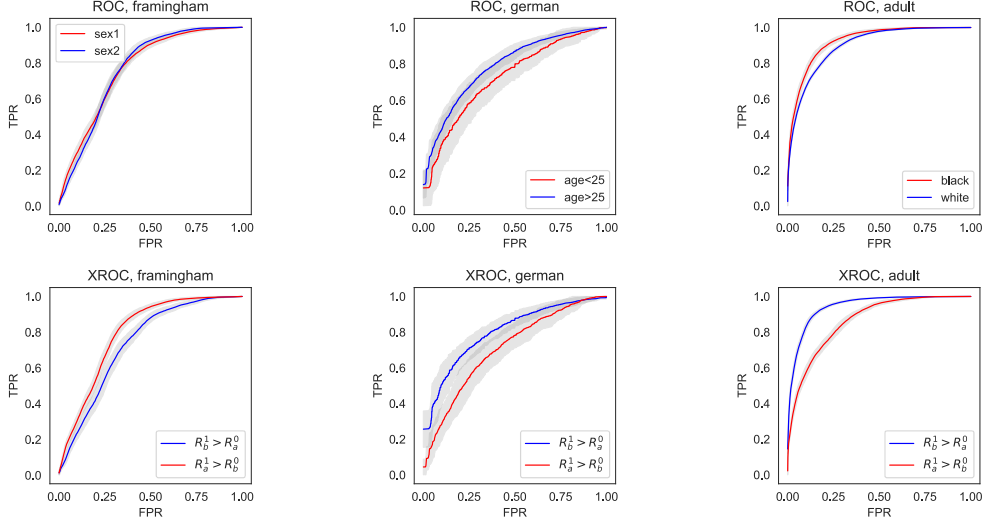

(a) Framingham: $a = $ Non-F., $b = $ Female, $1 = $ CHD

(b) German: $a = $ Age $< 25$, $b = $ Age $\geq 25$, $0 = $ Default

(c) Adult: $a = $ Black, $b = $ White, $0 = $ Low income

Figure 3: ROC and xROC curve comparison for Framingham, German credit, and Adult datasets.

## 6   Properties of the xAUC metric and Discussion

We proceed to characterize the xAUC metric and its interpretations as a measure of cross-group ranking accuracy. Notably, the xROC and xAUC implicitly compare performances of thresholds that are the same for different levels of the sensitive attribute, a restriction which tends to hold in applications under legal constraints regulating disparate treatment.

Next we point out that for a perfect classifier with $\mathrm{AUC} = 1$, the xAUC metrics are also 1. And, for a classifier that classifies completely at random achieving $\mathrm{AUC} = 0.5$, the xAUCs are also $0.5$.

**Impact of Score Distribution.** To demonstrate how risk score distributions affects the xAUC, we consider an example where we assume normally distributed risk scores within each group and outcome condition; we can then express the AUC in closed form in terms of the cdf of the convolution of the score distributions. Let $R_y^a \sim N(\mu_{ay}, \sigma_{ay}^2)$ be drawn independently. Then the xAUC is closed-form:

$\mathrm{xAUC}(a,b) = \Pr[R_1^a > R_0^b] = \Phi\left(\frac{\mu_{a1}-\mu_{b0}}{\sqrt{\sigma_{a1}^2+\sigma_{b0}^2}}\right)$. We may expect that $\mu_{a1} > \mu_{b0}$, in which case

$\Pr[R_1^a > R_0^b] > 0.5$. For fixed mean difference $\mu_{a1} - \mu_{b0}$ between the $a$-guilty and $b$-innocent (*e.g.*, in the COMPAS example), a decrease in either variance increases $\mathrm{xAUC}(a,b)$. For fixed variances, an increase in the separation between $a$-guilty and $b$-innocent $\mu_{a1} - \mu_{b0}$ increases $\mathrm{xAUC}(a,b)$. The

xAUC discrepancy is similarly closed form: $\mathrm{xAUC}(a,b) = \Phi\left(\frac{\mu_{a1}-\mu_{b0}}{\sqrt{\sigma_{a1}^2+\sigma_{b0}^2}}\right) - \Phi\left(\frac{\mu_{b1}-\mu_{a0}}{\sqrt{\sigma_{a0}^2+\sigma_{b1}^2}}\right)$. If

all variances are equal, then we will have a positive disparity (*i.e.*, in disfavor of $b$) if $\mu_{a1} - \mu_{b0} > \mu_{b1} - \mu_{a0}$ (and recall we generally expect both of these to be positive). This occurs if the separation between the advantaged-guilty and disadvantaged-innocent is smaller than the separation between the disadvantaged-guilty and advantaged-innocent. Alternatively, it occurs if $\mu_{a1} + \mu_{a0} > \mu_{b1} + \mu_{b0}$ so the overall mean scores of the disadvantaged are lower. If they are in fact equal, $\mu_{a1} + \mu_{a0} = \mu_{b1} + \mu_{b0}$ and $\mu_{a1} - \mu_{b0} > 0$, then we have a positive disparity whenever $\sigma_{a0}^2 - \sigma_{a1}^2 > \sigma_{b0}^2 - \sigma_{b1}^2$, that is, when in the $b$ class the difference in precision for innocents vs guilty is smaller than in group $a$. That is, disparate precision leads to xAUC disparities even with equal mean scores. In Appendix A.1 we include a toy example to illustrate a setting where the within-group AUCs remain the same but the xAUCs diverge.

Note that the xAUC metric compares probabilities of misranking errors *conditional* on drawing instances from either $Y = 0$ or $Y = 1$ distribution. When base rates differ, interpreting this disparity as normatively problematic implicitly assumes *equipoise* in that we want random individuals drawn

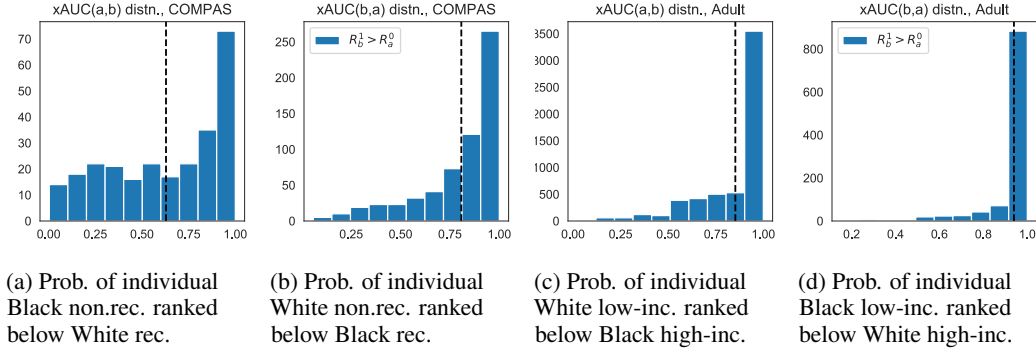

(a) Prob. of individual Black non.rec. ranked below White rec.

(b) Prob. of individual White non.rec. ranked below Black rec.

(c) Prob. of individual White low-inc. ranked below Black high-inc.

(d) Prob. of individual Black low-inc. ranked below White high-inc.

Figure 4: Distribution of conditional xAUCs for COMPAS and Adult datasets

with equal probability from the white innocent/black innocent populations to face similar misranking risks, not drawn from the population distribution of offending.

**Utility Allocation Interpretation.** When risk scores direct the expenditure of resources or benefits, we may interpret xAUC disparities as informative of group-level downstream utility disparities, if we expect beneficial resource or utility prioritizations which are *monotonic* in the score $R$. In particular, allowing for any monotonic allocation $u$, the xAUC measures $\Pr[u(R_1^a) > u(R_0^b)]$. Disparities in this measure suggest greater probability of confusion in terms of less effective utility allocation between the positive and negative classes of different groups. This property can be summarized by the integral representation of the xAUC disparities (*e.g.*, as in [34]) as differences between the *average rank* of positive examples from one group above negative examples from another group: $\Delta\text{xAUC} = \mathbb{E}_{R_1^a}\left[F_0^b(R_1^a)\right] - \mathbb{E}_{R_1^b}\left[F_0^a(R_1^b)\right]$.

**Diagnostics: Conditional xAUCs.** In addition to xAUC and xROC analysis, we consider the distribution of *conditional* xAUC ranking accuracies,

$$\text{xAUC}(a, b; R_b^0) := \mathbb{P}[R_a^1 > R_b^0 \mid R_b^0].$$

First note that $\text{xAUC}(a, b) = \mathbb{E}\left[\text{xAUC}(a, b; R_b^0)\right]$. Hence, this quantity is interpreted as the individual discrepancy faced by the $b$-innocent, the average of which over individuals gives the group disparity. We illustrate the histogram of $\text{xAUC}(a, b; R_b^0)$ probabilities over the individuals $R_b^0$ of the $A = b, Y = 0$ partition (and vice versa for $\text{xAUC}(b, a)$). For example, for COMPAS, we compute: how many white recidivators is this black non-recidivator correctly ranked above? xAUC is the average of these conditional accuracies, but the variance of this distribution is also informative of the range of misranking risk and of effect on individuals. We include these diagnostics in Fig. 4 and indicate the marginal xAUC with a black dotted line. For example, the first pair of plots for COMPAS illustrates that while the $\text{xAUC}(b, a)$ distribution of misranking errors faced by black recidivators appears to have light tails, such that the model is more accurate at ranking white non-recidivators above black recidivators, there is extensive probability mass for the $\text{xAUC}(a, b)$ distribution, even at the tails: there are 15 white recidivators who are misranked above nearly all black non-recidivators. Assessing the distribution of conditional xAUC can inform strategies for model improvement (such as those discussed in [10]) by directing attention to extreme error.

**The question of adjustment.** It is not immediately obvious that adjustment is an appropriate strategy for *fair risk scores* for downstream decision support, considering well-studied impossibility results for fair classification [13, 31]. For the sake of comparison to the literature on adjustment for fair classification such as [25], we discuss post-processing risk scores in Appendix E.1 and provide algorithms for equalizing xAUC. Adjustments from the fairness in ranking literature may not be suitable for risk scores: the method of [41] requires randomization over the space of rankers.

## 7 Conclusion

We emphasize that xAUC and xROC analysis is intended to diagnose potential issues with a model, in particular when summarizing model performance without fixed thresholds. The xROC curve and xAUC metrics provide insight on the disparities that may occur with the implementation of a predictive risk score in broader, but practically relevant settings, beyond binary classification.

## Acknowledgements

This material is based upon work supported by the National Science Foundation under Grant No. 1846210. This research was funded in part by JPMorgan Chase & Co. Any views or opinions expressed herein are solely those of the authors listed, and may differ from the views and opinions expressed by JPMorgan Chase & Co. or its affiliates. This material is not a product of the Research Department of J.P. Morgan Securities LLC. This material should not be construed as an individual recommendation for any particular client and is not intended as a recommendation of particular securities, financial instruments or strategies for a particular client. This material does not constitute a solicitation or offer in any jurisdiction.

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
