[Supplementary Material]

## A Analysis

*Proof of probabilistic derivation of the* xAUC. For the sake of completeness we include the probabilistic derivation of the xAUC, analogous to similar arguments for AUC [22, 23].

By a change of variables and observing that $\frac{d}{dv}G^{-1}(v) = -\frac{1}{G'(G^{-1}(v))} = \frac{1}{-f}$, if we consider the mapping between threshold $s$ that achieves TPR $v$, $s = G^{-1}(v)$, we can rewrite the AUC integrated over the space of *scores* s as

$$\int_{-\infty}^{\infty} G_1^a(s) f_0^b(s) ds$$

Recalling the conditional score distributions $R_1^a = R \mid Y = 1, A = a$ and $R_0^b = R \mid Y = 0, A = b$, then the probabilistic interpretation of the AUC follows by observing

$$\int_{-\infty}^{\infty} G_1^a(s) f_0^b(s) ds = \int_0^1 \Pr[R > s \mid Y = 1, A = a] \Pr[R = s \mid Y = 0] \, ds$$

$$= \int_0^1 \left( \int_0^1 \mathbb{I}(R_1^a > s) \Pr[R_1^a = t] dt \right) \Pr[R_0^b = s] \, ds$$

$$= \int_0^1 \int_0^1 \mathbb{I}(R_1^a > R_0^b) f_1^a(t) f_0^b(s) \, ds \, dt = \mathbb{E}[\mathbb{I}(R_1^a > R_0^b)] = \Pr[\mathbb{I}(R_1^a > R_0^b)]$$

$\square$

*Proof of Proposition 1.* We show this for the decomposition $\Pr[R_1 > R_0] = \sum_{a' \in \mathcal{A}} \Pr[A = a' \mid Y = 0] \Pr[R_1 > R_0^{a'}]$; the others follow by applying the same argument.

$$\sum_{a' \in \mathcal{A}} \Pr[A = a' \mid Y = 1] \Pr[R_1^{a'} > R_0] = \sum_{a' \in \mathcal{A}} \Pr[A = a' \mid Y = 1] \int_r \Pr[R > r \mid A = a', Y = 1] \Pr[R_0 = r] dr$$

$$= \sum_{a' \in \mathcal{A}} \int_r Pr[R > r, A = a' \mid Y = 1] \Pr[R_0 = r] dr$$

$$= \int_r \sum_{a' \in \mathcal{A}} Pr[R > r, A = a' \mid Y = 1] \Pr[R_0 = r] dr$$

$$= \int_r Pr[R > r \mid Y = 1] \Pr[R_0 = r] dr = \Pr[R_1 > R_0]$$

$\square$

### A.1 Example: same AUCs, different xAUCs

Again, for the sake of example, we assume normally distributed risk scores within each group and outcome condition and re-express the AUC in terms of the cdf of the convolution of the score distributions. For $R_0^a \sim N(\mu_{a0}, \sigma_{a0}^2)$, $R_1^b \sim N(\mu_{b1}, \sigma_{b1}^2)$, (drawn independently, conditional on outcome $Y = 1, Y = 0$), the xAUC is closed-form, $\Pr[R_1^a > R_0^b] = \Phi\left(\frac{\mu_{b0} - \mu_{a1}}{\sqrt{\sigma_{a1}^2 + \sigma_{b0}^2}}\right)$. To further gain intuition, we consider settings where the score distributions have equivalent within-group xAUC scores, and what parameters yield xAUC disparities.

$$\max\left\{ \left| \Phi\left(\frac{\mu_{a0} - \mu_{b1}}{\sqrt{\sigma_{b1}^2 + \sigma_{a0}^2}}\right) - \Phi\left(\frac{\mu_{b0} - \mu_{a1}}{\sqrt{\sigma_{b0}^2 + \sigma_{a1}^2}}\right) \right| : \frac{\mu_{a0} - \mu_{a1}}{\sqrt{\sigma_{a1}^2 + \sigma_{a0}^2}} = \frac{\mu_{b0} - \mu_{b1}}{\sqrt{\sigma_{b1}^2 + \sigma_{b0}^2}} \right\}$$

For the sake of concreteness we fix scalars for the parameters of group $a$. We then vary group $b$ parameters. The constraint of equal AUCs corresponds to the level curve $\sigma_{a1}^2 + \sigma_{a0}^2 = \sigma_{b1}^2 + \sigma_{b0}^2$. Let $\sigma_{a1}^2, \sigma_{a0}^2 = 0.25$, and $\mu_{a0} = 0.25$, $\mu_{a1} = 0.75$. We consider constraints $\mu \in [0, 1], \sigma^2 \leq 0.5$ to approximate densities on $[0, 1]$ Assume nontrivial classification performance corresponds with $\mu_1 > 0.5, \mu_1 < 0.5$ (the distribution is suitably peaked).

Then the remaining d.o.f. on the parameters are those for the $A = b$ group:

$$\max \left| \Phi \left( \frac{0.25 - \mu_{b1}}{\sqrt{\sigma_{b1}^2 + 0.25}} \right) - \Phi \left( \frac{\mu_{b0} - 0.75}{\sqrt{\sigma_{b0}^2 + 0.25}} \right) \right|$$
$$\text{s.t. } 0.5(\sigma_{b1}^2 + \sigma_{b0}^2) = (\mu_{b0} - \mu_{b1})^2$$

If we fix variances, $\sigma_{b1}^2, \sigma_{b0}^2 = 0.5$, then this disparity depends only on the means, and we can maximize the disparity by letting $\mu_{b0} \to 0, \mu_{b1} \to 1$. Otherwise if we fix the mean disparity, again we achieve maximal disparity by $\sigma_{b1} \to 0$ and $\sigma_{b0} \to 0.5$ (or vice versa).

# B  Additional Empirics

## B.1  Balanced xROC curves and score distributions

(a) Balanced xROC curves for Framingham ($a, b$ for female, male)

(b) KDEs of outcome- and group-conditional score distributions

(c) Balanced xROC curves for German ($a, b$ for black, white)

(d) KDEs of outcome- and group-conditional score distributions

(e) Balanced xROC curves for Adult ($a, b$ for black, white)

(f) KDEs of outcome- and group-conditional score distributions

Figure 5: Comparison of balanced xROC curves for Framingham, German, and Adult datasets

We compute the similar xROC decomposition for all datasets. For Framingham and German, the balanced XROC decompositions do not suggest unequal ranking disparity burden on the innocent or guilty class in particular. For the Adult dataset, the $\text{xAUC}_0$ disparity is higher than the $\text{xAUC}_1$ disparity, suggesting that the misranking disparity is incurred by low-income whites who are spuriously recognized as high-income (and therefore might be disproportionately extended economic opportunity via *e.g.* favorable loan terms). The Framingham data is obtained from http://biostat.mc.vanderbilt.edu/DataSets.

Framingham, German, and Adult have more peaked distributions (more certain) for the $Y = 0$ class with more uniform distributions for the $Y = 1$ class; the adult income dataset exhibits the greatest contrast in variance between the $Y = 0$ and $Y = 1$ class.

## C  Standard errors for reported metrics

Table 2: Standard errors of the metrics (AUC, xAUC, Brier scores for calibration) for different datasets.

|  |  | COMPAS | | Framingham | | German | | Adult | |
|---|---|---|---|---|---|---|---|---|---|
|  | $A =$ | $a$ | $b$ | $a$ | $b$ | $a$ | $b$ | $a$ | $b$ |
| Log Reg. | AUC | 0.011 | 0.018 | 0.016 | 0.014 | 0.049 | 0.029 | 0.007 | 0.004 |
|  | Brier | 0.004 | 0.006 | 0.007 | 0.006 | 0.023 | 0.012 | 0.004 | 0.002 |
|  | XAUC | 0.023 | 0.018 | 0.013 | 0.02 | 0.048 | 0.031 | 0.01 | 0.004 |
|  | $\text{XAUC}^1$ | 0.012 | 0.015 | 0.012 | 0.014 | 0.044 | 0.024 | 0.009 | 0.003 |
|  | $\text{XAUC}^0$ | 0.012 | 0.019 | 0.015 | 0.012 | 0.032 | 0.029 | 0.004 | 0.004 |
| RankBoost cal. | AUC | 0.011 | 0.014 | 0.015 | 0.013 | 0.045 | 0.027 | 0.008 | 0.003 |
|  | Brier | 0.004 | 0.005 | 0.005 | 0.004 | 0.022 | 0.009 | 0.003 | 0.002 |
|  | XAUC | 0.025 | 0.016 | 0.012 | 0.017 | 0.044 | 0.031 | 0.01 | 0.004 |
|  | $\text{XAUC}^1$ | 0.012 | 0.013 | 0.012 | 0.013 | 0.041 | 0.024 | 0.009 | 0.003 |
|  | $\text{XAUC}^0$ | 0.011 | 0.019 | 0.014 | 0.01 | 0.03 | 0.026 | 0.004 | 0.003 |

## D  Reproducibility checklist

- Data preprocessing and exclusion: We use the preprocessed datasets from the repository of [20] for COMPAS, German, and Adult, and all of the available data from the Framingham study.
- Train/validation/test: We train models on a 70% data split and evaluate xAUC, AUC and ROC, xROC on a 30% out of sample split.
- Hyper-parameters: we use sklearn defaults for the assessed methods.
- Evaluation runs: 50.
- Computing infrastructure: MacBook Pro, 16gb RAM.
- Further discussion on exact evaluation approach in Sec. 5

## E  xAUC postprocessing adjustment

| (a) COMPAS | (b) Framingham | (c) German | (d) Adult |

Figure 6: XROC curves, before and after adjustment

### E.1  Adjusting Scores for Equal xAUC

We study the possibility of post-processing adjustments of a predicted risk score that yield equal xAUC across groups, noting that the exact nature of the problem domain may pose strong barriers

Figure 7: TPR and FPR curves over thresholds ($G_y^a$), and adjusted curves for group $A = a$ ($\tilde{G}_y^a$)

Table 3: Metrics before and after xAUC parametric adjustment

|  | COMPAS | Fram. | German | Adult |
|---|---|---|---|---|
| AUC (original) | 0.743 | 0.771 | 0.798 | 0.905 |
| AUC (adjusted) | 0.730 | 0.772 | 0.779 | 0.902 |
| $\alpha^*$ | 4.70 | 3.20 | 4.71 | 4.43 |
| xAUC$(a,b)$ | 0.724 | 0.761 | 0.753 | 0.895 |
| xAUC$(b,a)$ | 0.716 | 0.758 | 0.760 | 0.898 |

to the implementability or individual fairness properties of post-processing adjustment. The results are intended to illustrate the distortionary extent that would be required to achieve equal xAUC by preprocessing.

Without loss of generality, we consider transformations $h : \mathbb{R} \mapsto \mathbb{R}$ on group $b$. When $h$ is monotonic, the within-group AUC is preserved.

$$\Pr[h(R_1^b) - R_0^a > 0] = \Pr[R_1^a - h(R_0^b) > 0]$$
$$= \int G_1^b(h^{-1}((G_0^a)^{-1}(s)))ds = \int G_1^a(h((G_0^b)^{-1}(s)))ds$$

Although solving analytically for the fixed point is difficult, empirically, we can simply optimize the xAUC disparity over parametrized classes of monotonic transformations $h$, such as the logistic transformation $h(\alpha, \beta) = \frac{1}{1+\exp(-(\alpha x+\beta))}$. We can further restrict the strength of transformation by restricting the range of parameters.

In Fig. 6 we plot the unadjusted and adjusted xROC curves (dashed) resulting from a transformation which equalizes the xAUC; we transform group $A = a$, the disadvantaged group. We optimize the empirical xAUC disparity over the space of parameters $\alpha \in [0, 5]$, fixing the offset $b = -2$. In Fig. 7, we plot the complementary cdfs $G_y^a$ corresponding to evaluating TPRs and FPRs over thresholds, as well as for the adjusted score (red). In table 3, we show the optimal parameters achieving the lowest xAUC disparity, which occurs with relatively little impact on the population AUC, although it reduces the xAUC$(b, a)$ of the advantaged group.

## E.2 Fair classification post-processing and the xAUC disparity

One might consider applying the post-processing adjustment of Hardt et al. [24], implementing the group-specific thresholds as group-specific shifts to the score distribution. Note that an equalized odds adjustment would equalize the TPR/FPR behavior for every threshold; since equalized odds might require randomization between two thresholds, there is no monotonic transform that equalizes the xROC curves for every thresholds.

We instead consider the reduction in xAUC disparity from applying the "equality of opportunity" adjustment that only equalizes TPR. For any specified true positive rate $\rho$, consider group-specific thresholds $\theta_a, \theta_b$ achieving $\rho$. These thresholds satisfy that $G_1^a(\theta_a) = G_1^b(\theta_b)$. Then $\theta_b = (G_1^b)^{-1}(G_1^a(\theta_a))$. The score transformation on $R$ that achieves equal TPRs is:

$$h(r, A) = \begin{cases} r & \text{if } A = a \\ (G_1^a)^{-1}(G_1^b(r)) & \text{if } A = b \end{cases}$$

461 **Proposition 2.** *The corresponding xAUC under an equality of opportunity adjustment, where*
462 $\tilde{R}_{eqop} = h(R)$, *is:*

$$\Delta\mathrm{xAUC}(\tilde{R}_{eqop}) = \mathrm{AUC}^b - \mathrm{AUC}^a$$

*Proof.*

$$
\begin{aligned}
\Delta\mathrm{xAUC}(\tilde{R}_{eqop}) &= \int G_1^a \left((G_1^a)^{-1}(G_1^b((G_0^b)^{-1}(s)))\right) ds \\
&\quad - \int G_1^b \left((G_1^b)^{-1}(G_1^a((G_0^a)^{-1}(s)))\right) ds \\
&= \int (G_1^b((G_0^b)^{-1}(s))) ds - \int G_1^a((G_0^a)^{-1}(s)) ds \\
&= \mathrm{AUC}^b - \mathrm{AUC}^a
\end{aligned}
$$

463 □