[Reviews · NeurIPS 2019]

Reviewer 1



Specifically considering the setting where model provides a score that can then be used later in making a decision is novel to my knowledge. The paper is clearly written and introduces an original criterion that can also be generalised to non-binary decisions. Ultimate significance of this style of work remains uncertain - it is arguable that eventually such investigations must shift to the discussions of applicable fairness criteria in certain specific situations, e.g. credit scoring, video recommendation, search result ranking etc, with concrete guidance for practice.

Reviewer 2



This paper is a fairly straightforward but sensible extension of ROC/AUC to compare the quality of ranking across groups. xAUC is just the probability that a positive instance of group a is ranked above a negative instance of group b. The paper is well structured and clearly written and I would expect these metrics to be widely adopted in quantifying fairness. I have read the author response and comments from other reviewers. I am still of the opinion that this paper represents a significant contribution. I strongly argue that it be accepted. I agree that this paper does not tell you when you should sacrifice accuracy to reduce xAUC disparity. However, I think is unreasonable to expect that it answer that question, as such an answer will be incredibly context dependent and will be based more on sociology, political science & philosophy than on machine learning. Almost no paper on fairness in ML would have been published if this was the standard. I agree that there are too many papers in ML introducing new fairness metrics with very limited justification for them. But I don't think this paper falls in that category because: 1) They show how their metric helps clarify the Compas debate - which is a seminal example of fairness in ML, 2) Their metric is closely connected to (and a means for visualising) concerns relating to separation - which is one of the fundamental, widely discussed and used existing notions of fairness. From this point of view they are demonstrating a way of quantifying an existing fairness concept in the setting of ranking rather than introducing an entirely new (and disconnected metric). 3) This paper has done a substantially better job of clarifying the implications of their metric and how it connects with other metrics than most papers in this space. A clear understanding of the properties of a metric (as is given in Section 6) forms the starting point for the discussion of whether it is or not appropriate within a given context.

Reviewer 3



This paper is tough to review. On one hand, it's well written and carefully thought out. But I don't come away from the paper with a clear idea of what I should do with xAUC, or why I should prefer it over other measures. On occasion, the authors describe xAUC as measuring misranking, which -- if this is indeed what it measures -- would suggest an immediate intervention to remedy the misranking. However, the authors caution against efforts to adjust the scores to equalize xAUC. At other times xAUC is described as a diagnostic, but even the Bayes-optimal predictor could produce significant xAUC disparities, while a miscalibrated score (where there is clear misranking -- one group's risk is being consistently over or under-estimated) could produce no xAUC disparities. So it's unclear how one should interpret xAUC differences. The fairness literature is awash with fairness metrics, and I think the burden is on those proposing new metrics to make a compelling case for why we should prefer their metric to the existing alternatives. This paper has not convinced me that xAUC provides significant value over existing metrics. Response to author feedback: It's not the case that there are "no metrics specifically for disparate impacts of continuous risk probability scores." For example, in their paper "Risk, Race, & Recidivism: Predictive Bias and Disparate Impact," Skeem & Lowenkamp measure bias in continuous risk scores by fitting logistic regression curves to the score-outcome relationship for each group. Following the American Psychological Association's "Principles for the Validation and Use of Personnel Selection Procedures" they interpret significant differences in either slopes or intercepts between groups as evidence of bias in the risk scores (in other words, they require sufficiency to hold). This has been the APA's recommended way to measure bias in risk scores since at least 2003. COMPAS provides an illustrative example. xAUC suggests bias, while the APA's approach (applied by Flores et al. in their rejoinder to the ProPublica analysis titled "False Positives, False Negatives, and False Analyses") finds none. Which result should we believe? This paper doesn't make a compelling case for why the well-established approach should be discarded in favor of xAUC.

[Author Response · NeurIPS 2019]

1 We thank the reviewers for their feedback.

**Reviewer 1**

Thank you for the encouraging words. We have tried as much as possible to provide insight on what xAUC discrepancies mean and to provide diagnostic tools to inspect them in order to inform practice. We agree further work is needed to push this in specific applications. While this is beyond the scope of this present paper, we have employed xAUC/xROC metrics to assess fairness in more recent work and will continue to push this direction in application work.

**Reviewer 2**

We appreciate the kind words and your expectation for xAUC to be widely adopted. And, yes, thank you: we will mention the connection to separation (separation for a score implies but is strictly stronger than $\Delta \text{xAUC} = 0$; note also that $\Delta \text{xAUC}$ is a metric for assessing level of unfairness rather than just a criterion). Re code dependencies: we will update the checklist; we use Python 2.7 with numpy/pandas/sklearn/matplotlib/scipy.

**Reviewer 3**

"fairness literature is awash with fairness metrics [...] make a compelling case for why we should prefer their metric to the existing alternatives."

- Firstly, there are no metrics specifically for **disparate impacts of continuous risk probability scores**, as acknowledged by R1. We distinguish between conditional probability estimation with observed binary outcomes, and fair regression which studies parity constraints on loss with respect to observed regression outcomes (Agarwal et al, 2019; Zink and Rose 2019).

- We specifically featured the COMPAS example, a commonly studied dataset, to make a compelling case for this metric: attention to xAUC, its probabilistic interpretation, and its decompositions, illuminates causes of unfairness. We highlighted that previous arguments (e.g., Dieterich et al. [18]) which compared within-group AUCs did not provide a meaningful notion of accuracy equity in decision-support settings, so that **relative to such naive metrics that exist, xAUC should be preferred**. We further featured other datasets to demonstrate the breadth of applicability.

Re: base rates: In Proposition 1, we provide a decomposition that illustrates the connection between base rates, xAUC disparities, and marginal AUC. This shows that base rates are just one ingredient and disparities can exist even with equal base rates.

Re: what action should be taken after xAUC is measured:

- The legitimacy of direct post-training adjustment is highly contested both in the fairness literature and in practice and is highly context-dependent, and this is in no way specific to the xAUC. Part of this ambiguity relates to the overloaded use of "predictive risk scores" in the fairness literature, which our work in part seeks to clarify. We highlight that different interpretations may be more or less directly applicable based on the problem setting. If the setting does not admit adjustment (e.g. risk assessment in healthcare to inform patient decisions), we highlight work that discusses alternative approaches for directly improving performance, such as choice of covariates. When these avenues are available, it is unclear if adjustment is advisable.

- What action should be taken is highly context-dependent, and we illustrate how different contexts lead to different recommendations. Recognizing that the issue of direct adjustment (we believe) remains contested in the fairness literature and for practitioners, due to legal and Pareto-violation concerns, we (or anyone) wouldn't be able to provide an honest and realistic algorithmic solution to address xAUC disparities. See e.g. Hellman 2019, "Measuring Algorithmic Unfairness", which seeks to establish general principles for preferring calibration vs. impact-based or base-rate sensitive metrics.

- Nonetheless, we relate the idea of post-processing adjustment of scores to post-processing classifiers by adjusting thresholds. Furthermore, the results we include in the supplement do provide the tools for $\Delta \text{xAUC}$-minimizing adjustment of scores for those who would choose to do so.

[Meta-Review · NeurIPS 2019]

There was a lot of discussion over this paper. The central concern was this: * did the authors provide sufficient justification for the value of xAUC over other fairness measures * Is a good case made for the measure having normative value. After reading the reviews, the feedback and the discussion, it seems like 1. The authors are very careful to guardrail claims about the applicability of the measure - they in fact do NOT claim that this measure should be viewed as superior to other measures, but merely that it provides a context that can't otherwise be easily seen. That argument is compelling. 2. Reviewers correctly called out the measure as having issues when base rates differ (as many measures do), and to be fair the authors do acknowledge this in the paper, although they don't directly address it. I think this is a potential weakness of the measure, but I don't view it as fatal. Overall, the xAUC Measure does add an extra dimension to the discussions, and separately, I think the idea of adapting AUC to group fairness measures to be quite elegant.